# Uni- and Multivariate Analyses of Cancer Risk in Cytologically Indeterminate Thyroid Nodules: A Single-Center Experience

**DOI:** 10.3390/cancers16050875

**Published:** 2024-02-22

**Authors:** Enrico Battistella, Marica Mirabella, Luca Pomba, Riccardo Toniato, Francesca Giacomini, Giovanna Magni, Antonio Toniato

**Affiliations:** 1Endocrine Surgery Unit, Department of Surgery, Veneto Institute of Oncology, IOV-IRCCS, Via Gattamelata 64, 35128 Padua, Italy; marica.mirabella@iov.veneto.it (M.M.); luca.pomba@iov.veneto.it (L.P.); francesca.giacomini@iov.veneto.it (F.G.); antonio.toniato@iov.veneto.it (A.T.); 2School of Medicine, University of Padua, Via Giustiniani 2, 35128 Padua, Italy; riccardo.toniato@studenti.unipd.it; 3Clinical Research Unit, Veneto Institute of Oncology, IOV-IRCCS, Via Gattamelata 64, 35128 Padua, Italy; giovanna.magni@iov.veneto.it

**Keywords:** thyroid nodules, cytologically indeterminate nodules, thyroid cancer, thyroid surgery, thyroid malignancy, endocrine surgery

## Abstract

**Simple Summary:**

Almost 30% of thyroid nodules are cytologically indeterminate (TIR3A/3B) and the risk of malignancy reported in the literature for TIR3A thyroid nodules ranges from 5% to 15% and for TIR3B from 15% to 30%. In our retrospective monocenter study, we have shown how these percentages are higher in clinical practice and that a tailored treatment should be performed considering the clinical characteristics of each patient.

**Abstract:**

Every year in Italy, about 60,000 new cases of nodular thyroid pathology are diagnosed, of which almost 30% are cytologically indeterminate (TIR3A/3B). The risk of malignancy reported in the literature on thyroid nodules ranges from 5% to 15% for TIR3A and from 15% to 30% for TIR3B. It is suspected that these percentages are higher in practice. We performed univariate and multivariate analyses of clinical risk factors. The medical records of 291 patients who underwent surgery for cytologically indeterminate nodular thyroid disease were retrospectively reviewed. Clinical parameters and preoperative serum markers were then compared between the benign nodular thyroid disease and thyroid cancer groups. For each patient, clinical characteristics, comorbidities, neck ultrasonographic features, and histological reports were statistically analyzed using Chi-squared and Fisher’s exact tests. A total of 134 malignant neoplasms were found (46%), divided into 55 cases (35%) in the TIR3A group and 79 cases (59%) in the TIR3B group. Statistical analysis was not significant in both populations for both sex and age (TIR3A *p*-value = 0.5097 and *p*-value = 0.1430, TIR3B *p*-value = 0.5191 *p*-value = 0.3384), while it was statistically significant in patients with TIR3A nodules associated with thyroiditis (*p*-value = 0.0009). In addition, the patients with TIR3A and 3B nodules were stratified by ultrasound risk for the prediction of malignancy and it was significant (*p* = 0.0004 and *p* < 0.0001). In light of these results, it emerges that surgical treatment of nodular thyroid pathology with indeterminate cytology TIR3A should always be considered, and surgery for TIR3B is mandatory.

## 1. Introduction

Thyroid nodules are a common condition in clinical practice, especially in countries where iodine prophylaxis is not practiced. Every year in Italy, about 60,000 new cases of nodular thyroid pathology are diagnosed, of which 5–7% are malignant in nature. It can present as a single lesion or in a multinodular form, in the context of a normal gland or a goiter [1,2,3,4,5]. Ultrasound is the gold standard diagnostic tool for nodular pathology because it allows for the definition and standardization of the characteristics of thyroid nodules to stratify their risk of malignancy and to establish a comparison of them over time [6]. Fine-needle aspiration cytology (FNAC) is a minimally invasive, echo-guided diagnostic procedure that allows the definition of the cytological diagnosis of the benign or malignant nature of a thyroid nodule that is suspicious on ultrasound. It is the gold standard for diagnosing malignant neoplasms because of its high sensitivity (83%) and specificity (92%). FNAC allows for the selection of patients for conservative management or surgical intervention [7]. The Italian classification of thyroid cytology used in this study is the one currently in use, the SIAEPEC-IAP Italian Classification 2014 [8]. In the present study, we have considered patients who have undergone thyroid surgery with a TIR3 lesion (subdivided into 3A, indeterminate low-risk lesion, and 3B, indeterminate high-risk lesion) in the last 9 years at the U.O.C. of Endocrine Surgery of the Veneto Oncology Institute. In particular, the aim of this study is to verify the incidence of malignant thyroid neoplasms found on histological examination, which, according to the scientific literature, is around 5–15% and 15–30%, respectively [9]. It is suspected that these percentages are higher in practice. In addition, observation of the identified case series will allow us to describe the demographic, clinical, and ultrasonographic characteristics of thyroid nodules that are predominantly associated with the presence of a malignant neoplasm.

## 2. Materials and Methods

### 2.1. Patients and Study Design

All adult patients with nodular pathology with indeterminate cytology who underwent hemithyroidectomy or total thyroidectomy at our center between 2015 and March 2023 were included in this retrospective observational study.

For each patient, the following characteristics were included in the analysis: demographics (age and sex), presence of thyroiditis associated with the nodular pathology, ultrasonographic characteristics of the nodule, stratified for risk according to the 2016 AACE-AME classification (low, intermediate, and high risk of malignancy), and surgery of choice (thyroidectomy or hemithyroidectomy).

In addition, for those with malignancy, the stage of the disease as determined by histological examination was reported according to TNM staging. Therefore, in cases of hemithyroidectomy with subsequent thyroid totalization, radiotherapy with I-131 or reintervention was described.

### 2.2. Statistical Analysis

Data were described overall and stratified by nodule classification: TIR3A and TIR3B. Chi-squared and Fisher’s exact tests were used to test for differences between the two nodule types. The calculated tests were considered statistically significant when *p* < 0.05. The following software was used: SAS software, version 9.4 (SAS Institute Inc., SAS Campus Drive, Cary, NC, USA).

To meet the main objective of determining the incidence of subjects with observed malignancy, the total number of subjects with malignancy was related to the total number of subjects observed. This percentage was accompanied by a 95% confidence interval.

Similarly, the incidence was calculated within the subjects with TIR3A and TIR3B nodules only, with a 95% confidence interval.

### 2.3. Ethical Aspects

This study was conducted in accordance with Good Clinical Practice (GCP) guidelines and the Declaration of Helsinki on human experimentation. It was also conducted in accordance with local regulations and guidelines. This retrospective observational study was approved by the Ethics Committee of the Veneto Oncology Institute IOV-IRCCS on 26 July 2023 (IOV-ENDCH-01-2023).

## 3. Results

From 2015 to March 2023, 2203 patients underwent thyroid surgery, of whom 291 patients (13.2%) underwent surgery for thyroid nodules with indeterminate cytology TIR3. Of these, 157 were TIR3A nodules (54%) and 134 were TIR3B nodules (46%). The mean age of the patients who underwent surgery was 53.1 years (minimum 19 years to maximum 85 years), divided by sex as follows: 225 females (77.3%) and 66 males (22.7%). Thyroiditis was found in 71 patients with thyroid nodules (24.4%), particularly in 25 patients with TIR3A nodules (15.9%) and in 46 patients with TIR3B nodules (34.3%).

Patients were also stratified according to the ultrasonographic characteristics of the nodule using the AACE-AME 2016 classification. For TIR3A nodules, 18 patients had a low risk of malignancy (12.3%), 106 patients had an intermediate risk (72.6%), and 22 patients had a high risk (15.1%). For TIR3B nodules, 10 patients had a low risk of malignancy (7.9%), 85 patients had an intermediate risk (66.9%), and 32 patients had a high risk (25.2%).

Molecular analysis was performed in 14 patients from the TIR3A group without significant results: we recorded three k-RAS mutations and one BRAFV600E. The other ten patients did not have any mutations.

The type of surgery performed was hemithyroidectomy in 59 patients, or 20.3% of the total [44 TIR3A (28%), 15 TIR3B (11.2%)], and total thyroidectomy in 232 patients, or 79.7% of the total [113 TIR3A (72%), 119 TIR3B (88.8%)]. Furthermore, sampling of the central compartment was performed in 115 patients (39.5%), especially when lymph nodes results enlarged due to the co-existence of thyroiditis (Table 1).

Histological examination revealed a total of 134 malignant neoplasms (46%) divided into 55 cases (35%) in the TIR3A group and 79 cases (59%) in the TIR3B group (Table 2).

Papillary thyroid carcinoma was found in 114 cases (85.9%) in the histological report subdivided as follows: 49 cases (31.2%) of papillary carcinoma among TIR3A nodules, and 66 cases (49.2%) among TIR3B nodules. In addition, the histological report indicated follicular carcinoma in 18 cases: 6 cases (3.8%) for TIR3A nodules and 12 cases (9%) for TIR3B nodules. One case of TIR3B was a thyroid metastasis from breast carcinoma (0.8%); however, all other cases were non-neoplastic nodules (157 cases, 54%). Among the nodules found to be malignant, the TNM staging was analyzed: two patients (3.6%) in the TIR3A group had central lymph node metastases (N1a), while the number was eight patients (10.3%) in the TIR3B group. In this group, four patients (5.1%) had retropharyngeal lymph node metastases (N1b).

The stage of the disease was also assessed and was found to be stage I in all TIR3A nodules, stage I in 91.1% (71 cases) of TIR3B nodules, stage II in 7.7% (6 cases), and stage III in 1.2% (1 case).

Postoperative treatment with radioablative therapy (I-131) for malignant pathology was performed in 18 cases (32.7%) in patients with TIR3A nodules and in 52 cases (66.7%) in patients with TIR3B nodules.

Thyroid totalization reintervention was necessary in three cases, two cases of TIR3A (5.5%) and one case of TIR3B (3.8%) (Table 3).

### Statistical Analysis

From the statistical analysis conducted, there were no statistically significant differences between the patients with benign and malignant histology in the two populations for both sex and age (TIR3A *p*-value = 0.5097 and *p*-value = 0.1430, TIR3B *p*-value = 0.5191 *p*-value = 0.3384).

The analysis was statistically significant in patients with TIR3A nodules associated with thyroiditis for predicting malignancy (*p*-value = 0.0009). However, in subjects with TIR3B nodules, the presence of thyroiditis did not appear to be associated with the presence of malignancy (*p*-value = 0.1512) (Table 4).

Continuing the analysis, the ability to predict malignancy was evaluated by stratifying the patients with TIR3A and 3B nodules by ultrasound risk. The probability of the nodule being malignant increased with increasing ultrasound risk in both groups (Cochran–Armitage trend test: *p* = 0.0004, Cochran–Armitage trend test: *p* < 0.0001). Table 5.

The median follow-up is 45 months, and all patients are still alive. Two patients (2.5%) with TIR3B nodules have undergone latero-cervical lymphadenectomy for the recurrence of papillary thyroid carcinoma. The other six patients have thyroglobulin above the normal range without evidence of disease.

## 4. Discussion

The prevalence of thyroid nodules in the general population is high—up to 60% as documented by high-resolution ultrasonography. Thyroid cancer is relatively rare, with only approximately 16 new cases diagnosed per 100,000 adults per year in the United States, while 28.2 per 100,000 females and 10 per 100,000 males per year are diagnosed in Italy [1,3,10,11]. Given that most thyroid nodules are benign, it is beneficial to distinguish preoperatively between nodules that are likely to be benign and those that are likely to be malignant in order to minimize unnecessary surgery for benign nodules and to reserve surgery for clinically significant malignancies. The diagnostic–therapeutic approach to thyroid nodules with indeterminate cytology (TIR3A and TIR3B) remains a challenge in thyroid pathology. Our study investigated and analyzed factors associated with the risk of developing carcinoma in thyroid nodules with indeterminate cytology. In the literature, clinical factors that generally increase the risk of malignant nodules are age (<20 or >70 years); male sex; previous exposure to radiation to the head and neck region; family history of benign, malignant, or related genetic thyroid disorders with thyroid tumors (MEN syndromes, Cowden syndrome, Carney complex, familial adenomatous polyposis); mode of onset; rate of growth; and presence of compressive symptoms (cough, dyspnea, dysphonia, dysphagia, sensation of constriction) [12]. According to the studies published to date, the relationship between Hashimoto’s thyroiditis and papillary thyroid carcinoma remains unclear. Therefore, this issue needs to be addressed in the future through large prospective studies [13,14]. Ultrasonographic evaluation of the thyroid nodule is another key element in the stratification of the risk of malignancy. In clinical practice, as well as in this study, the system used is the 2016 AACE-AME classification [15,16], which allows simplified ultrasound malignancy risk stratification into three classes (1, 2, and 3, representing low, intermediate, and high malignancy risk, respectively), compared with other classifications proposed by the ACR, ATA, and KTA [17,18,19,20,21], which are more complex and less directly applicable in clinical practice.

Molecular testing is an increasingly utilized adjunct in the evaluation of indeterminate thyroid nodules to avoid unnecessary surgical or diagnostic risk to the patient.

As reported in the 2023 European Thyroid Association Clinical Practice Guidelines for thyroid nodule management, molecular testing may improve diagnostic outcomes for thyroid nodules by identifying patients with indeterminate cytology as most likely benign using an integrated approach combining clinical anamnesis of the patients, ultrasonographic and cytological malignancy risk assessment with local outcome and test performance data. Currently, molecular FNA testing of indeterminate FNA outside of the USA is limited to mainly research use of local laboratory-developed tests and one funded test. Possible reasons for this include differences in health care systems, issues related to the lack of independent validation studies, the lack of long-term outcome studies for ‘benign’ molecular tests, and the high cost that currently limits their use outside the USA [22,23,24,25]. In our study, molecular testing was only performed in 14 patients with TIR3A nodules due to the high cost, and the results were not significant enough to report.

Our study showed that the risk rate of malignant pathology is 35% for TIR3A thyroid nodules and 59% for TIR3B. The risk of malignancy reported in the scientific literature ranges from 5 to 15% for TIR3A thyroid nodules and 15 to 30% for TIR3B, according to the 2021 AIOM Guidelines [9]. A 2018 meta-analysis conducted by the Italian Consensus for the Classification and Reporting of Thyroid Cytology [25] showed that the risk of malignant pathology was 17% for TIR3A nodules and 47% for TIR3B, confirming the reliability of the cytopathological classification of the 2014 Italian Consensus [8].

The indications described in the literature for TIR3A cytological nodules are close clinical and ultrasonographic follow-up, with repeat thyroid needle aspiration at subsequent follow-up, provided there are no risk factors suggesting surgical intervention [7]. For TIR3B nodules, surgical intervention is the priority [7]. In patients with clinical or ultrasonographic suspicion, surgical treatment with thyroid hemithyroidectomy is indicated, especially for patients with solitary nodules with indeterminate cytology [25,26,27] and for nodular lesions smaller than 4 cm [28]. Total thyroidectomy may be considered depending on the clinical context, the coexistence of contralateral nodules, the presence of thyroiditis, and patient preference [7]. In our study, 59 patients underwent hemithyroidectomy, 20.3% of the total [44 TIR3A (28%), 15 TIR3B (11.2%)], and 232 patients underwent total thyroidectomy, or 79.7% of the total [113 TIR3A (72%), 119 TIR3B (88.8%)]. Thyroid totalization reintervention was necessary in three cases, both for TIR3A (5.5%) and for TIR3B (3.8%).

Other anamnestic variables for the risk of malignant pathology were taken into account, such as age, sex, and the presence of Hashimoto’s thyroiditis in the population in question and ultrasonographic characteristics of the thyroid nodule with indeterminate cytology, stratified according to the 2016 AACE-AME ultrasound classification. According to the available data and evaluable anamnesis, it was found that there was no statistically significant difference between patients with benign and malignant histology in the two populations for both sex and for age (TIR3A *p*-value = 0.5097 and *p*-value = 0.1430, TIR3B *p*-value = 0.5191 *p*-value = 0.3384); moreover, in patients with thyroiditis, it was confirmed that there was a higher presence of malignancy, especially in subjects with TIR3A nodules (*p*-value = 0.0009).

In subjects with TIR3B nodules, the presence of thyroiditis does not appear to be associated with the presence of malignant pathology (*p*-value = 0.1512). Finally, with regard to ultrasound risk, stratified according to the 2016 AACE-AME classification [15], it was observed that in the TIR3A group, as the ultrasound risk increases, the probability that the nodule will be malignant increases (Cochran–Armitage trend test: *p* = 0.0004), as well as in the TIR3B nodule group (Cochran–Armitage trend test: *p* < 0.0001) [29,30].

The limitations of this single-center, retrospective study are that it considers a population of patients in whom surgical treatment was indicated by the endocrinologist (hemithyroidectomy or total thyroidectomy) and does not include a population under active surveillance. As a retrospective study, the anamnestic data collected and consistently reported correspond to those included in the analysis (age, sex, presence of thyroiditis, and ultrasound characteristics of the thyroid nodules TIR3A and TIR3B).

## 5. Conclusions

In this study, factors associated with the risk of developing thyroid carcinoma in nodules with indeterminate cytology were found in 35% of TIR3A (55 cases) and 59% of TIR3B (79 cases) nodules. The coexistence of thyroiditis and TIR3A nodules was a significant risk factor in predicting malignancy. Furthermore, the probability of the nodule being malignant increased with increasing ultrasound risk in both groups.

Considering these results, it emerges that surgical treatment of nodular thyroid pathology with indeterminate cytology TIR3A should be considered in the case of two consecutive TIR3A FNAC reports, coexistence of thyroiditis, increasing size of the thyroid nodule, changing ultrasonographic characteristics of the nodule, and appearance of nodule-related symptoms. Surgery should be limited to a hemithyroidectomy if there are no other indications for total thyroidectomy.

Indeterminate TIR3B nodules must be treated with surgery: hemithyroidectomy or total thyroidectomy, depending on the size of the nodule, the presence of multinodular disease, and the coexistence of thyroiditis, and the patient’s wishes.

Retrospective single-center studies and new prospective and multi-center studies are needed to strengthen the conclusions and consequent therapeutic hypotheses of this study.

## Figures and Tables

**Table 1 cancers-16-00875-t001:** Clinical features of patients enrolled in this retrospective study.

	TIR		
	3A	3B	All
**Median age**	55.2 years	50.6 years	53.1 years
**Female**	119 (75.8%)	106 (79.1%)	225 (77.3%)
**Male**	38 (24.2%)	28 (20.9)	66 (22.7%)
**Thyroiditis**	25 (15.9%)	46 (34.4%)	71 (24.4%)
**US low risk**	18 (12.3%)	10 (7.9%)	28 (10.2%)
**US intermediate risk**	106 (72.6%)	85 (66.9%)	191 (70%)
**US high risk**	22 (15.1%)	32 (25.2%)	54 (19.8%)
**Hemithyroidectomy**	44 (28%)	113 (72%)	157 (100%)
**Total thyroidectomy**	15 (11.2%)	119 (88.8%)	134 (100%)
**Malignancy rate**	55 (35%)	79 (59%)	134 (46%)

**Table 2 cancers-16-00875-t002:** Incidence of malignancy of cytologically indeterminate thyroid nodules in our analyses.

	No. of Patients	Incidence of Malignancy	Interval of Confidence
**TIR3A**	157	35%	27.6–43.0
**TIR3B**	134	59%	50.1–67.4
**All**	291	46%	38.4–53

**Table 3 cancers-16-00875-t003:** Histopathological characteristics of our retrospective study.

	TIR		
	3A	3B	All
**Papillary thyroid** **carcinoma**	49 (89.1%)	66 (83.6%)	114 (85.9%)
**Follicular thyroid** **carcinoma**	6 (10.9%)	12 (15.2%)	18 (13.4%)
**Thyroid metastases**	/	1 (1.3%)	1 (0.7%)
**TNM stage I**	55 (100%)	71(91.1%)	125 (94.7%)
**TNM stage II**	/	6 (7.7%)	6 (4.5%)
**TNM stage III**	/	1 (1.2%)	1 (0.8%)
**Radioablative** **therapy**	18 (32.7%)	52 (66.7%)	70 (52.6%)

**Table 4 cancers-16-00875-t004:** Risk of malignancy in patients with co-existence of cytologically indeterminate thyroid nodules and thyroiditis (Chi-Square test, *p*-value< 0.01).

Thyroiditis	TIR											
	3a						3b					
	Histotype				All		Histotype					
	Benign Disease		Malignant Disease				Benign Disease		Malignant Disease			
	N	%	N	%	N	%	N	%	N	%	N	%
**NO**	93	91.2	39	70.9	132	84.1	40	72.7	48	60.8	88	65.7
**YES**	9	8.8	16	29.1	25	15.9	15	27.3	31	39.2	46	34.3
**Total**	102	100.0	55	100.0	157	100.0	55	100.0	79	100.0	134	100.0
**Chi-Square**	**0.0009**						0.1512					

**Table 5 cancers-16-00875-t005:** The probability of the nodule being malignant increased with increasing ultrasound risk in both groups (Cochran–Armitage trend test: *p* = 0.0004, Cochran–Armitage trend test: *p* < 0.0001). The ultrasound risk was not applicable in 18 patients.

AACE/AME 2016	TIR											
	3a						3b					
	Histotype				All		Histotype				All	
	Benign		Malignant				Benign		Malignant			
	N	%	N	%	N	%	N	%	N	%	N	%
**Low-risk thyroid lesion**	17	17.7	1	2.0	18	12.3	10	18.9	.	.	10	7.9
**Intermediate risk thyroid** **lesion**	70	72.9	36	72.0	106	72.6	40	75.5	45	60.8	85	66.9
**High-risk thyroid lesion**	9	9.4	13	26.0	22	15.1	3	5.7	29	39.2	32	25.2
**Total**	96	100.0	50	100.0	146	100.0	53	100.0	74	100.0	127	100.0
**Chi-Square**	**0.00018**						**<0.0001**					

## Data Availability

Data could be found in the patients’ medical records.

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
