# Peer review of "Uni- and Multivariate Analyses of Cancer Risk in Cytologically Indeterminate Thyroid Nodules: A Single-Center Experience"

_cancers, 2024, doi:10.3390/cancers16050875_

Round 1

Reviewer 1 Report

Comments and Suggestions for Authors

The manuscript titled "Uni- and multivariate analysis of cancer risk in cytologically indeterminate thyroid nodules: a single-center experience" addresses a highly relevant clinical issue. Unfortunately, the problem investigated in this research has already been explored in numerous studies.

However this research has serious flaws:

1.     The study is retrospective and conducted at a single center, potentially diminishing its statistical significance.

2.     The conclusions do not account for the very low risk of cancer in the group of patients classified as "US low risk of malignancy" according to the AACE-AME 2016 classification. This aspect should be emphasized in the conclusions since the study is not exclusively dedicated to surgeons.

Minor comment: There is an error in the spelling of the word "hemithyroidectomy" in Table 1.

Author Response

Answer to reviewer:

1-     The study is retrospective and conducted at a single center, potentially diminishing its statistical significance.

- We have declared this limitation of the study in the conclusion

2-  The conclusions do not account for the very low risk of cancer in the group of patients classified as "US low risk of malignancy" according to the AACE-AME 2016 classification. This aspect should be emphasized in the conclusions since the study is not exclusively dedicated to surgeons.

- We have emphasized as correctly suggested the results of the statistical analysis (thyroiditis and US risk stratification) in the conclusion section. 

Minor comment: There is an error in the spelling of the word "hemithyroidectomy" in Table 1.

Reviewer 2 Report

Comments and Suggestions for Authors

Manuscript Title: Uni- and multivariate analysis of cancer risk in cytologically in-determinate thyroid nodules: a single-center experience

. . . . . . . . . . . . . . . . . . . . . . . . . . . . . . . . . . . . . . . . . . . . . . . . . . . . . . . . . . . . . . . . . . . . . . . . . . . . . . . . . . . . . .

Criticisms of the current manuscript study on univariate and multivariate analysis of cancer risk in cytologically in-determinate thyroid nodules are presented below.

It would be more appropriate to present the material and method in stages with headings. herefore, the same applies to the results section of the manuscript. It would be more appropriate to correct this disorganized situation that makes it difficult for the reader to follow the manuscript.

It is suggested that the term misspelled as "EMITHHYROIDECTOMY" in Table 1 be corrected to the correct spelling "HEMITHYROIDECTOMY".

p-values should be indicated where relevant in the table. This can be a sign etc.

. . . . . . . . . . . . . . . . . . . . . . . . . . . . . . . . . . . . . . . . . . . . . . . . . . . . . . . . . . . . . . . . . . . . . . . . . . . . . . . . . . . . . .

Author Response

Answer to reviewer

1- It would be more appropriate to present the material and method in stages with headings. herefore, the same applies to the results section of the manuscript. It would be more appropriate to correct this disorganized situation that makes it difficult for the reader to follow the manuscript.

As suggested, material and methods section and the results section are divided in subchapter to be more clear. 

2- It is suggested that the term misspelled as "EMITHHYROIDECTOMY" in Table 1 be corrected to the correct spelling "HEMITHYROIDECTOMY".

- It is corrected

3- p-values should be indicated where relevant in the table. This can be a sign etc.

The relevant results were underlined as suggested in the table.

Reviewer 3 Report

Comments and Suggestions for Authors

 Uni- and multivariate analysis of cancer risk in cytologically in-2 determinate thyroid nodules: a single-center experience.

The article presented by the authors has some clinical relevance with regard to the treatment of thyroid nodules whose cytology is inconclusive, demonstrating that their clinical reality has different values to what is described in the literature. Although I have enjoyed reading the article, which I always find useful, the results obtained do not really add value to the knowledge that already exists.

Possible suggestions/alterations for the authors:

- The authors should review the bibliographical research that supports the introduction, improving the state of the art by introducing some more recent articles on the topic.

- As far as the bibliography is concerned, they should be referenced according to the journal's rules, always put them in the same way and add the Doi of the articles.

- Whenever it appears in the text: e.g. ....goiter. [1] should be ....goiter [1].   

                                                                     ….intervention. [3] deve estar assim …. Intervention [3].

- Results:

- line 114: the results described in this paragraph are not shown in table 2 but in table 1

- line 116: This paragraph of the results should be revised and rewritten. You don't understand the % in the text and when you look and compare it with table 3, it becomes confusing. You could add another way of reading the table and add the percentages. In line 118 where it says ...16 cases it should read ...18 cases.

- In table 5, replace Medium risk thyroid lesion with Intermediate risk thyroid lesion to be consistent with the text.

- The paragraph on line 134 and line 138 could be moved to line 122 as it follows on from the reading of table 3.

- Although the molecular results were not significant, they could have been.

-  The discussion is well organized and in line with the findings

Author Response

Answer to reviewer

  1. line 114: the results described in this paragraph are not shown in table 2 but in table

- We added a sentences in line 114 to clarify the speech and in table 2

2- line 116: This paragraph of the results should be revised and rewritten. You don't understand the % in the text and when you look and compare it with table 3, it becomes confusing. You could add another way of reading the table and add the percentages. In line 118 where it says ...16 cases it should read ...18 cases.

-We have rewritten the linee 116: "Papillary thyroid carcinoma was found in 114 cases (85,9%) in the histological report subdivided as follows: 49 cases (31.2%) of papillary carcinoma among TIR3A nodules, and 66 cases (49.2%) among TIR3B nodules.". Line 118 was corrected.

3- In table 5, replace Medium risk thyroid lesion with Intermediate risk thyroid lesion to be consistent with the text.

- Intermediate risk thyroid lesion was corrected in the table 5

4- The paragraph on line 134 and line 138 could be moved to line 122 as it follows on from the reading of table 3.

- This change was performed as suggested

5- Although the molecular results were not significant, they could have been.

- We added our results

-  The discussion is well organized and in line with the findings

Round 2

Reviewer 1 Report

Comments and Suggestions for Authors

Thank you, I accept the manuscript in this version.

Reviewer 2 Report

Comments and Suggestions for Authors

It was observed that the author's made the suggested corrections for the manuscript.